# Genome-Wide Expression Profile in People with Optic Neuritis Associated with Multiple Sclerosis

**DOI:** 10.3390/biomedicines11082209

**Published:** 2023-08-07

**Authors:** Mario Habek, Antonela Blazekovic, Kristina Gotovac Jercic, Nela Pivac, Tiago Fleming Outero, Fran Borovecki, Vesna Brinar

**Affiliations:** 1Department of Neurology, Referral Center for Autonomic Nervous System Disorders, University Hospital Centre Zagreb, 10000 Zagreb, Croatia; 2Department for Functional Genomics, Center for Translational and Clinical Research, University of Zagreb School of Medicine, University Hospital Centre Zagreb, 10000 Zagreb, Croatia; 3Department for Anatomy and Clinical Anatomy, University of Zagreb School of Medicine, 10000 Zagreb, Croatia; 4Department of Neurology, University Hospital Centre Zagreb, 10000 Zagreb, Croatia; 5Division of Molecular Medicine, Rudjer Bošković Institute, 10002 Zagreb, Croatia; 6Department of Experimental Neurodegeneration, Centre for Biostructural Imaging of Neurodegeneration, University Medical Centre Göttingen, 37075 Göttingen, Germany; 7Max Planck Institute for Experimental Medicine, 37075 Göttingen, Germany; 8Translational and Clinical Research Institute, Faculty of Medical Sciences, Newcastle University, Framlington Place, Newcastle upon Tyne NE1 7RU, UK; 9German Centre for Neurodegenerative Diseases (DZNE), 17475 Göttingen, Germany

**Keywords:** optic neuritis, multiple sclerosis, genome-wide expression analysis

## Abstract

The aim of this study was to perform a genome-wide expression analysis of whole-blood samples from people with optic neuritis (ON) and to determine differentially expressed mRNAs compared to healthy control subjects. The study included eight people with acute ON and six healthy control subjects. Gene expression was analyzed using DNA microarrays for whole-human-genome analysis, which contain 54,675 25-base pairs. The additional biostatistical analysis included gene ontology analysis and gene set enrichment analysis (GSEA). Quantitative RT-PCR (qPCR) was used to confirm selected differentially expressed genes. In total, 722 differently expressed genes were identified, with 377 exhibiting increased, and 345 decreased, expression. Gene ontology analysis and GSEA revealed that protein phosphorylation and intracellular compartment, apoptosis inhibition, pathways involved in cell cycles, T and B cell functions, and anti-inflammatory central nervous system (CNS) pathways are implicated in ON pathology. qPCR confirmed the differential expression of eight selected genes, with *SLPI*, *CR3*, and *ITGA4* exhibiting statistically significant results. In conclusion, whole-blood gene expression analysis showed significant differences in the expression profiles of people with ON compared to healthy control subjects. Additionally, pathways involved in T cell regulation and anti-inflammatory pathways within CNS were identified as important in the early phases of MS.

## 1. Introduction

Multiple sclerosis (MS) is a clinically heterogeneous disease characterized by widespread inflammatory demyelinating plaques that can involve various parts of the central nervous system, including optic nerves. To date, the etiology and pathogenesis of MS have not been fully elucidated, but with the advent of new research and the discovery of new molecules, the landscape of MS understanding is rapidly changing. The inflammatory demyelination of the optic nerve is called optic neuritis (ON) and is a possible first presentation of MS [1]. Although some never experience further neurological symptoms, 85% of people with an acute episode of neurological deficit consistent with one or more demyelinating lesions will develop MS [2]. It is estimated that ON is present in the initial phase in 20–40% of people with MS (pwMS) and that up to 80% of pwMS will experience ON at some point during the disease [3,4]. On the other hand, the Optic Neuritis Treatment Trial found that MS developed in half of the people with ON (pwON) within 15 years [5].

Several genome-wide expression studies in pwMS for diagnostic, prognostic, or therapeutic purposes have been performed. One of the first studies showed different expression in 34 among the 4000 tested genes in pwMS compared to healthy control subjects [6]. Subsequent studies expanded the number of differentially expressed genes, but also provided some novel mechanisms or confirmed existing ones already known to be involved in MS pathogenesis [7,8]. Additionally, the expression profiling approach was used in monitoring treatment response in several studies, providing us with biomarkers that can be used for immunomodulatory treatment optimization [9,10]. Research into the genomic basis of ON is limited, and there are not many data on genetic factors as potential biomarkers of ON [11]. However, there are some studies indicating an involvement of polymorphisms of genes *CYP4F2* (cytochrome P450 family 4 subfamily F member 2) and *MMP-2* (matrix metalloproteinase-2) in ON development [12,13,14,15].

The aim of this study was to determine changes in mRNA expression in blood samples of pwON compared to healthy control subjects. 

## 2. Materials and Methods

### 2.1. Patient’s Selection

Peripheral blood samples were collected from 8 acute pwON within 4 weeks from symptom onset and 6 age- and gender-matched healthy control subjects. The neurological status of pwON was determined using the Expanded Disability Status Scale performed by an experienced MS neurologist. All pwON had the following tests performed: brain and spinal cord MRI (magnetic resonance imaging), CSF (cerebrospinal fluid) analysis (cell count, protein level, and presence of oligoclonal IgG bands), immunological tests (extractable nuclear antigens (ENAs), antinuclear antibody (ANA), Anti-neutrophil cytoplasmic antibodies (ANCAs), and IgG and IgM anticardiolipin antibodies (aCLs)), and IgM and IgG antibodies against *Borrelia burgdorferi* in serum and CSF. The study participants were recruited at the Department for Neurology, University Hospital Centre Zagreb, Croatia. The control subjects included individuals without any abnormal findings in the tests above and without vision problems. Written informed consent was obtained from all participants before conducting any procedure related to this study. All procedures were carried out according to the regulations of the Declaration of Helsinki. The study was approved by the Institutional Ethics Committee University of School of Medicine (8.1-19/299-2).

### 2.2. RNA Isolation and Gene Profiling

All samples were collected in the morning hours. Total RNA was extracted from whole blood using the PAXgene blood RNA kit (PaxGene, Preanalytix, QIAGEN N.V. and Becton, Dickinson and Company (BD), Heidelberg, Germany) according to the manufacturer’s instructions. All samples were treated with the RNase-free DNase set (Qiagen). The concentration of RNA was determined by measuring the absorbance at 260 nm (A260) in a spectrophotometer (Eppendorf, Hamburg, Germany), and the quality of all total RNA samples was analyzed using a 2100 Bioanalyzer (Agilent Technologies, Santa Clara, CA, USA). 

Gene expression was analyzed with DNA chips for the analysis of the whole human genome (Human Genome U133 PLUS 2.0 GeneChip, Affymetrix, CA, USA) which contains 54,675 25-probe sets, according to the manufacturer’s protocol. All chips were run in the Department for functional genomics, University of Zagreb, School of Medicine. 

The following software programs were used for data analysis: Affymetrix Gene Chip Command Console Software (AGCC) 4.0, S-plus, R, and DChip. AGCC 4.0 was used for determining gene expression levels of the given gene on the Affymetrix Human Genome U133 PLUS 2.0 gene chips. Standard quality measures and normalization for the Affymetrix GeneChip (3′/5′ ratios and trimmed mean normalization) were used in the experiments. The genes with low expression levels have lower copy numbers of mRNA and are most susceptible to technical noise. Therefore, in the analysis of the Affymetrix microarray data, only genes with “signal” intensity in at least one sample above the “target intensity” of 100 were considered for further analysis. All parameters of quality control measures are presented in Appendix A. The list of the significantly changed genes was further filtered with the following parameters: “signal” intensity in at least one sample above the “target intensity” of 100, an expression ratio of average ON/average healthy control > 1.8 or <0.6 as cut-off values, and *p* value < 0.01. All arrays were run in the same core facility at the Department for Functional Genomics, Center for Translational and Clinical Research, University of Zagreb School of Medicine.

### 2.3. Gene Ontology Analysis

In order to classify the significantly differentially expressed genes according to their gene ontology properties, “Database for Annotation, Visualization, and Integrated Discovery” (DAVID) was used. By using the “Functional Annotation Clustering” tool in DAVID gene clusters, significantly similar ontologies were identified and then tested across the whole gene list. 

### 2.4. Analysis of Significantly Altered Pathways

To determine significantly altered functional and signaling pathways, we processed the results of gene chip expression with the GSEA (Gene Set Enrichment Analysis) 2.0 software package. We used the GSEA-P software package and a set of 522 predefined gene groups. The primary statistical method used for the investigation of GSEA was normalized enrichment score (NES) which reflects the degree to which a certain group of genes is overrepresented at the top or the bottom of the gene list. The second statistical method used was false discovery rate (FDR), which represents the estimated probability that a group of genes with the default NES includes a false-positive finding. As an additional indicator of statistical significance, we used nominal *p*-value (NOM *p*-value).

### 2.5. Pathway Analysis Using the Ingenuity Pathway Analysis-IPA 

To ascertain the functional pathways which may be regulating differential expression in ON, we performed pathway analysis using the Ingenuity Pathway Analysis-IPA (Ingenuity^®^ Systems, www.ingenuity.com; accessed on 1 September 2022). The IPA was performed by uploading the aforementioned microarray data into the commercially available IPA software v.7.6 for biological function and pathway analysis. The IPA “Core Analysis” was performed using the Ingenuity Knowledge Base (Genes only) as the reference set, using direct and indirect relationships for network analysis and data from mammal species and all tissues, cell lines, and data sources.

### 2.6. Gene Expression Analysis Using Real-Time PCR

Quantitative RT-PCR (qPCR) was performed using 2 µg of the total RNA isolated from 9 pwON and 6 healthy controls. The RNA was transcribed using the Superscript First-Strand Synthesis System (Invitrogen Life Technologies, Carlsbad, CA, USA) with random hexamer primers according to the manufacturer’s protocol. Gene expressions of interest were measured using the LightCycler FastStart DNA Master SYBR Green kit (Roche Diagnostics, Mannheim, Germany) on the LightCycler instrument (Roche Diagnostics, Mannheim, Germany). Results were represented as fold change in comparative expression level. The sequences of primers are shown in Table 1.

## 3. Results

### 3.1. Microarray Analysis of Global Gene Expression Changes in the Blood of pwON 

Using Affymetrix GeneChip U133A, we analyzed global gene expression changes in blood samples from eight pwON and six healthy, age- and gender-matched control subjects. All pwON had at least one demyelinating lesion present on the brain MRI and positive oligoclonal IgG bands. All other immunological tests and Borrelia burgdorfery antibodies were negative. The Affymetrix platform identified 722 significantly changed genes (*p* < 0.05, average ON/average healthy controls expression ratio >1.8 or <0.6, expression level > 100). Of the 722 significantly changed genes, 377 were up-regulated and 345 were down-regulated (Figure 1).

Among the differentially expressed mRNAs, genes pertaining to immune response, such as receptor type C for protein tyrosine phosphatase; genes belonging to the solute carrier family, such as *SLC11A1* (solute carrier family 11 member 1); and genes involved in transcription, such as *MXD1* (MAX dimerization protein 1), showed statistically increased expression. Among the control samples, genes involved in the cell cycle, such as *CDKN1C* (cyclin-dependent kinase inhibitor 1C); genes important for immune response, such as *CCR3* (chemokine (C-C motif) receptor 3); and genes involved in membrane functioning, such as *SLAMF1* (Signaling Lymphocytic Activation Molecule Family Member 7), were among the most significantly upregulated.

In order to compare the obtained expression profiles with previously published studies, we reanalyzed the dataset obtained within the study by Kempinnen et al. [16]. Using the same statistical parameters as in our study, we found that out of the 722 genes with differential expression identified in our study, 280 also exhibited differential expression in the aforementioned study (Appendix A). Additionally, three of the genes subsequently chosen for qPCR confirmation were among the 280 differentially expressed genes, namely *PTPRC* (protein tyrosine phosphatase receptor type C), *ITGA4* (integrin subunit alpha 4), and *CCR3*. More importantly, the study by Kampinnen et al. also tried to conduct a systematic review of seven microarray studies in order to identify overlapping differentially expressed genes. Using the list of 229 genes found to be differentially expressed in at least two out of the seven studies, we identified 7 genes that were present in our dataset. Most importantly, out of the genes subsequently confirmed by qPCR in our study, *ITGA4* was differentially expressed in previous studies as well.

### 3.2. Gene Ontology Analysis

To determine the gene ontology terms that were mostly represented in the group of significantly changed genes, we used the statistical software DAVID. We separately analyzed the groups of up- and-down-regulated genes. In the up-regulated group of genes, the analysis showed a significant representation of terms related to phosphorylated proteins, protein binding, alternative splicing, posttranslational modifications, cell death, and apoptosis. In the down-regulated group of genes, the analysis showed a significant representation of terms related to the negative regulation of programmed cell death, anti-apoptosis, the negative regulation of apoptosis, the regulation of cell transport, and the cell cycle.

In order to analyze not only the representation of certain ontological terms but also their mutual functional relationship, we performed Gene Ontology Functional Annotation Clustering. In the set of up-regulated genes, functional clustering showed three ontological term groups with an enrichment factor greater than 2. The first group included genes involved in protein binding, binding, and the cell nucleus; the second group included genes involved in protein phosphorylation, phosphor metabolism, amino acid phosphorylation, protein kinase activity, protein metabolism, phosphorylation, a serine–threonine kinase, catalytic activity, and transferase activity; and the third group included genes involved in cytoskeleton function, actin binding, the organization and biogenesis of the cytoskeleton, and parts of the cytoskeleton and organelles that are not membrane-bound. In the set of down-regulated genes, functional clustering showed three ontological terms groups with an enrichment factor greater than 2. The first group included genes involved in the negative regulation of cell death, negative regulation of apoptosis, negative regulation of biological processes, cell development, programmed death, and cell differentiation; the second group included genes involved in the regulation of the cell cycle; and the third group included genes involved in immunoglobulin metabolism. 

### 3.3. Gene Set Enrichment Analysis

The total number of enriched pathways in pwON was 534, with 4 functional groups of genes enriched with FDR < 0.25 and 45 with a nominal *p*-value < 0.05. The total number of functional groups with NES > 1 in the pwON group was 240. In the healthy controls, the number of enriched pathways was 913, 114 of which had a nominal *p*-value < 0.05 and 486 had NES > 1. Five gene sets were significantly associated with: the interleukin 7 *(IL7*) pathway (Appendix A), interleukin 2 receptor beta (*IL2RB*) pathway, interleukin 2 receptor alpha (*IL2RA*) pathway, interleukin 4 (*IL4*) pathway (Appendix A), and interleukin 4 receptor (*IL4R*) pathway (Appendix A).

### 3.4. Ingenuity Pathway Analysis

To correlate expression findings with relevant biological processes and pathways, we performed pathway analysis using the manually curated literature-based Ingenuity database. The top ten canonical pathways over-represented among our differentially expressed genes were iCOS-iCOSL signaling in T helper cells (9 molecules), Nur77 signaling in T lymphocytes (6 molecules), CTLA4 (Cytotoxic T-Lymphocyte-Associated Protein 4) signaling in cytotoxic T lymphocytes (8 molecules), CD28 signaling in T helper cells (9 molecules), TREM1 (Triggering Receptor Expressed On Myeloid Cells 1) signaling (6 molecules), graft-versus-host disease signaling (5 molecules), autoimmune thyroid disease signaling (4 molecules), nitric oxide signaling in the cardiovascular system (6 molecules), integrin signaling (11 molecules), and altered T cell and B cell signaling in rheumatoid arthritis (6 molecules). 

Among the top five canonical networks activated within the differentially expressed gene set were cell-to-cell signaling and interaction, cellular compromise, inflammatory response, immunological disease, and inflammatory disease, cell-to-cell signaling and cell death (Figure 2).

The figure depicts the interrelations of the genes belonging to the cell death signaling pathway. The graphic illustrates the molecular relationships between genes involved in cell death. Molecules are represented as nodes, and the biological relationship between two nodes is represented as a line. All edges are supported by at least one reference from the literature, from a textbook, or from canonical information stored in the Ingenuity Knowledge Base. The intensity of the node color indicates the degree of up- or down-regulation. The green color depicts decreased expression, while the red color depicts increased expression in the ON samples when compared to the controls. Nodes are displayed using various shapes that represent the functional class of the gene product.

In order to identify a subset of genes that could usefully classify pwON and healthy controls, 722 genes were ranked according to their *p* values, highest fold change, highest expression levels, and consistency of fold change in each individual ON sample compared to their age and gender-matched control. Probe sequences that corresponded to duplicates of the same genes, as well as probes for unknown or hypothetical proteins, were removed. In the case of duplicate entries, probes with the most statistically significant changes were used in further analysis. Using such criteria, we selected eight representative genes to be further tested with qPCR: Protein tyrosine phosphatase, receptor type, C (*PTPRC*), Secretory leukocyte peptidase inhibitor (*SLPI*), Leupaxin (*LPXN*), Chemokine (C-C motif) receptor 3 (*CCR3*), Integrin, alpha 4 (antigen CD49D, alpha 4 subunits of VLA-4 receptor) (*ITGA4*), CD28 molecule (*CD28*), and *SLAMF7*. The genes were selected according to the individual statistical significance of differential expression, but they were also chosen to represent significantly enriched gene sets and pathways derived from subsequent statistical analyses. Of the eight genes, all exhibited differential expression in concordance with that observed in microarray experiments, with three (*SLPI*, *CCR3*, and *ITGA4*) showing statistically significant differential mRNA expression (Table 2).

## 4. Discussion

Our results demonstrate a marked change in gene expression in the peripheral blood of pwON, with a total of 722 genes showing statistically significant differential expression. A similar count of altered genes was observed in pwMS in previous studies [8]. A recent analysis of a total of approximately 100,000 pwMS and control subjects revealed a detailed genetic landscape of common variations that could explain almost half of the heritability of the disease [17]. In addition to 200 autosomal genetic variants, this genome-wide map highlights another 416 associations that were replicated but did not reach the same level of statistical significance, suggesting numerous potential genetic links [17]. 

Furthermore, gene ontology analysis revealed a significant representation of terms related to protein phosphorylation and the intracellular compartment in the group of genes with increased expression in pwON included in this study. This may indicate an increased activation of blood cells and enhanced signaling activity since the activation of receptors and signaling molecules are often mediated through phosphorylation. These results are interesting in several aspects. It is believed that protein phosphorylation is a crucial link between the inflammatory and degenerative stages of MS. A currently prevailing opinion is that central nervous system (CNS)-confined inflammation in MS is associated with the release of soluble molecules, which can alter excitatory synaptic transmission and, finally, stimulate secondary neurodegenerative grey matter pathology [18]. Protein phosphorylation and metabolism play a key role in these processes. It is known that disorders related to optic neuritis are often associated with other autoimmune diseases. However, the causal relationship between these two conditions has not been fully studied. 

In our study, the increased cellular activity in whole-blood samples from pwON also showed a significant representation of terms related to the cytoskeleton and actin and protein binding. These findings speak in favor of early axonal damage in MS. In the CSF of pw MS compared with healthy subjects and patients with other neurological diseases, higher levels of the three major cytoskeletal neuronal proteins (actin, tubulin, and L-neurofilaments) were found [19]. A clear correlation between clinical disability measured using the Expanded Disability Status Scale (EDSS) was also observed. The potential involvement of programmed axon death in the pathogenesis of neurodegenerative diseases has been demonstrated in several studies [20]. Neurodegeneration occurs in the early stages of MS. However, the pathological mechanisms involved in neurodegeneration are not clear. This finding is particularly interesting because blocking programmed axonal death in animals has also been shown to be protective in several models of mitochondrial dysfunction disorders, including multiple sclerosis [21]. Furthermore, retinal ganglion cell degeneration has been shown to occur in the preclinical phase of optic neuritis. Herold et al. showed another aspect of the role of apoptosis in the development of optic neuritis. They hypothesized that the altered cleavage of amyloid precursor protein (APP) in neurons in the preclinical phase is associated with enhanced production of the intracellular domain of APP, which acts as a transcriptional regulator and, thus, initiates the apoptotic signaling cascade via the p53 gene [22]. 

On the other hand, the analysis of genes with reduced expression in pwON showed a significant representation of concepts related to the inhibition of apoptosis, the cell cycle, and the production of immunoglobulins. It has already been shown that, in MS, there is a genetic predisposition that leads to the inability of autoreactive immune cells to enter apoptosis [23]. Proteins that regulate apoptosis are abnormally expressed in active MS lesions, whereas analysis of peripheral blood gene microarrays has shown that a large percentage of genes that are affected by MS regulate apoptosis, as confirmed in this study [24]. Using an animal model of optic neuritis, Sindler et al. showed that inflammatory cell infiltration mediates demyelination and leads to direct axon injury [25]. The importance of cell cycle regulation in MS has been demonstrated in proteomic studies which revealed the expression dysregulation of proteins related to blood vessel development, cell structure, and cell cycle control [26]. The overall results of gene ontology analysis suggested an increased activity of lymphocytes in pw MS, with subsequently enhanced reactivity and high intracellular signaling. Given the apparent decrease in the representation of genes associated with the production of immunoglobulins, it can be assumed that the activation of T lymphocytes is a crucial process. These same cells also exhibit an inhibition of apoptosis and slowing of the cell cycle. This is consistent with research that has shown high levels of T helper (Th) cells and related cytokines and chemokines in lesions of the central nervous system and the cerebrospinal fluid of pwMS. It is hypothesized that they thereby contribute to the breakdown of the blood–brain barrier and ultimately result in neuroinflammation [27]. 

It is important to emphasize that the most significantly enriched signaling pathways and gene functional groups identified with GSEA and IPA in pwON are associated with the inflammatory response. Two genes that were identified as key genes carrying an increased risk of MS were *IL2R* and *IL7R* [28]. It has been shown that the *IL2RA* locus is associated with other autoimmune diseases like systemic lupus erythematosus and ANCA-associated systemic vasculitis [29]. The association of *IL2RA* locus with various autoimmune diseases suggests that the *IL2RA* pathway plays an important but still undefined role in each of these diseases. Disruption of the IL2R signaling pathway with a genetic defect in the gene for *IL2* or *IL2R* complex components results in severe T-cell-related autoimmunity, suggesting a key role of *ILR2* signaling in the development and function of regulatory T cells [30]. A systematic review and meta-analysis including 13,526 pwMS showed strong differences in blood *IL-2R* and *IL-23* levels between pwMS and control subjects [31]. Interleukin 7 is required for the development and survival of T lymphocytes. Once the T cells leave the thymus, *IL7* is essential for their survival, and the availability of *IL7* limits the number of T cells [32]. *IL7* also plays an important role in the development of B cells, including survival, proliferation, and maturation [32]. On the other hand, GSEA showed two pathways related to *IL4*. The regulation of CNS inflammation is essential to prevent irreversible cell damage that occurs in the neurodegenerative phase of MS. Studies on experimental autoimmune encephalomyelitis (EAE) have shown that the formation of *IL4* within the CNS is necessary to control autoimmune inflammation within the CNS [33]. In conclusion, GSEA and IPA analyses revealed the key role of pathways associated with the regulation function of T and B cells and pathways associated with anti-inflammatory processes within the CNS.

The qPCR analysis of eight representative genes, among others, showed higher secretory leukocyte protease inhibitor (*SLPI*) expression in pwON compared to healthy control subjects. *SLPI* consists of a protease inhibitor at the C-terminal domain and an antimicrobial at the N-terminal domain. Reviglio et al. showed that *SLPI* is not synthesized in eye tissues under normal physiological conditions [34]. *SLPI* exerts wide-ranging effects on inflammatory pathways and was up-regulated in an animal model of MS, but the biological role of *SLPI* in EAE was unknown [35]. One study performed by Müller et al. investigated whether the effects of *SLPI* on the immune system may have implications in diseases characterized pathologically by inflammation as a result of autoimmune mechanisms such as those in the MS [36]. By inducing SLPI-neutralizing antibodies in mice and rats, this study determined the clinical severity of the disease. In addition, the effects of *SLPI* on the anti-inflammatory cytokine TGF-b were studied. In EAE, *SLPI* exerts potent proinflammatory actions through the modulation of T cell activity, and its neutralization may be beneficial for the disease. Despite SLPI’s manifold anti-inflammatory properties, the severity of both rat and murine EAE was reduced by the induction of neutralizing SLPI antibodies. 

Another gene that was overexpressed in pwON was *PTPRC*. Although our *p*-value was slightly above statistical significance, this result is in concordance with previous studies [37,38]. Vidmar and colleagues demonstrated an increased variant burden among pwMS, predominantly among rare variants with high predicted pathogenicity. These variants were found in inflammasome genes (*NLRP1/3* and *CASP1*), genes mediating inflammasome inactivation via auto- and mitophagy (*RIPK2* and *MEFV*), and genes involved in the response to infection with DNA viruses (*POLR3A*, *DHX58*, and *IFIH1*) and type-1 interferons (*TYK2* and *PTPRC*) [38].

One of the main limitations of the study was the small number of included pwON, but it is important to note that the number of pwON at initial presentation and who did not receive treatment was not large. The majority of previous papers dealing with expression changes in the blood of patients with demyelinating diseases used mainly pwMS, which are easier to recruit. Nevertheless, even in those papers, the number of patients was not exceedingly high [7,8]. Another limitation is the fact that the analysis was limited to mRNA without an analysis of protein levels. 

## 5. Conclusions

In conclusion, the collective results point to the dysregulation of several key pathways associated with the regulation of T and B lymphocytes, leading to altered anti-inflammatory processes in the CNS. These data, confirmed using advanced statistical methods that ascertained the enrichment of functional gene sets and signaling pathways, suggest the existence of a definitive multi-directional relationship between CNS inflammation, T-cell-related autoimmunity, and cell death. This is the first whole-genome expression study of whole blood in pwON and points to the possible avenues of biomarker development for early-stage pwMS.

## Figures and Tables

**Figure 1 biomedicines-11-02209-f001:**
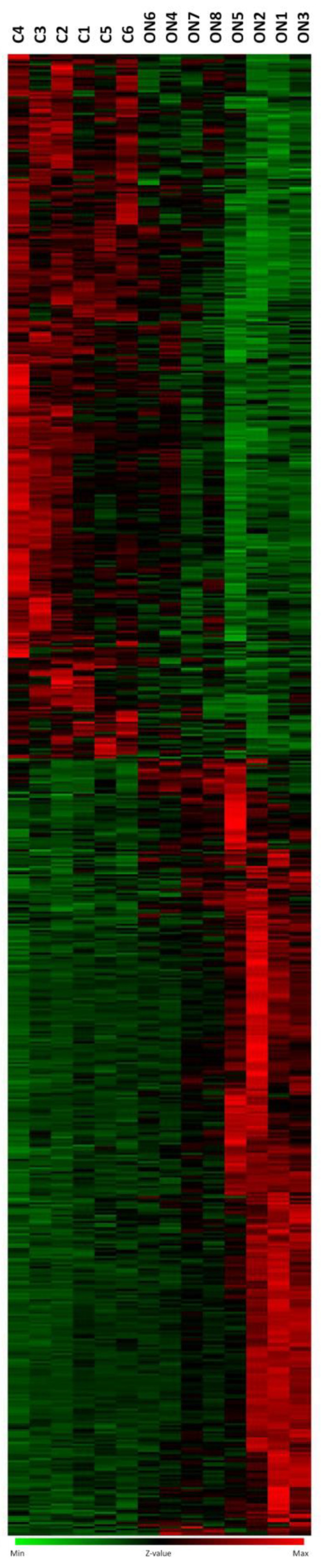
Gene expression profile in pwON and healthy controls. Genes were selected from 8 pwON and 6 healthy controls (*p* < 0.05, average ON/average healthy controls expression ratio >1.8 or <0.6, expression level > 100). Every column represents a sample, and the row represents a gene. The intensity of the color indicates the degree of up- or down-regulation. The green color depicts decreased expression, while red color depicts increased expression in the ON samples when compared to controls. (C1–C6—healthy controls, ON1–ON8—pwON.)

**Figure 2 biomedicines-11-02209-f002:**
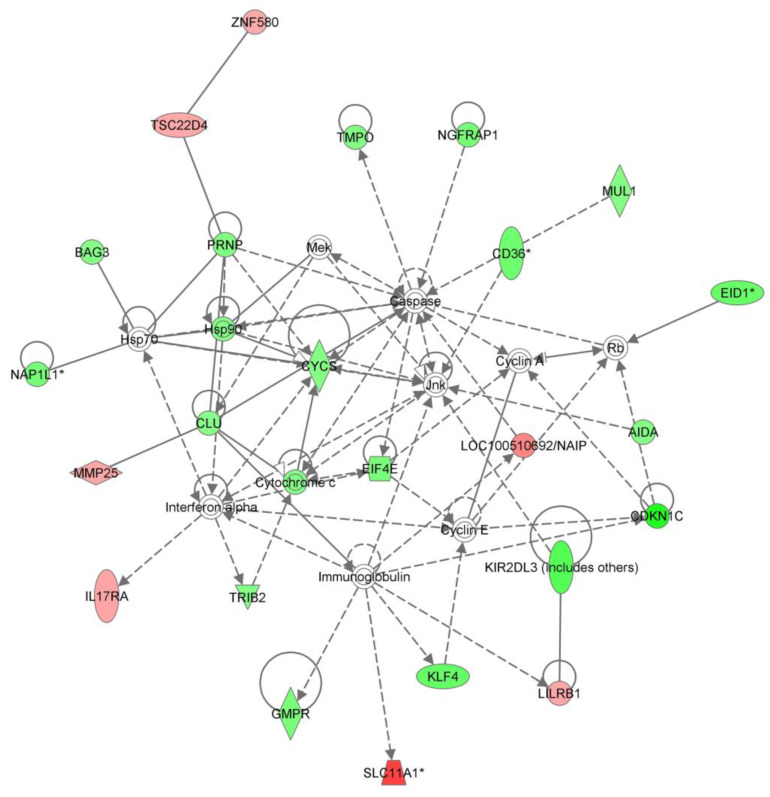
IPA analysis of the cell-death-related genes. * fuzzy search.

**Table 1 biomedicines-11-02209-t001:** Sequences of primers used in qPCR gene expression analysis.

Gene	5′ → 3′ Forward	5′ → 3′ Reverse
*PTPRC*	ACAGCCAGCACCTTTCCTAC	GTGCAGGTAAGGCAGCAGA
*SLC11A1*	GCCCTGTCCGTCTCCTTTATC	TGTTGGCACAGATGTTGAACG
*SLPI*	AGTGCCCAGTGACTTATGGC	CATGCCCATGCAACACTTCAA
*LPXN*	CTGCCAGAAACCGATTGCTG	ACTCCGCTCAAAGAAGGGACT
*CCR3*	ATACAGGAGGCTCCGAATTATGA	ATGCCCCCTGACATAGTGGAT
*ITGA4*	GCAGAGCACCATCAGAGAGG	GTCACTTCCAACGAGGTTTGTT
*CD28*	GCTTGCTAGTAACAGTGGCCT	GCGGGGAGTCATGTTCATGTA
*SLAMF7*	GAGCTGGTCGGTTCCGTTG	TGAAGGTCCAGACAATAGAGTCA

**Table 2 biomedicines-11-02209-t002:** Differential expression in a subset of genes selected for validation measured by qPCR.

Symbol	Accession No.	Description	Fold Change	*p*-Value *
*PTPRC*	NM_002838.4	Protein tyrosine phosphatase, receptor type, C	1.48	0.077
*SLPI*	NM_011414.3	Secretory leukocyte peptidase inhibitor	3.26	**0.001**
*LPXN*	NM_004811.2	Leupaxin	0.75	0.288
*CCR3*	NM_178329.2	Chemokine (C-C motif) receptor 3	0.41	**0.043**
*ITGA4*	NM_000885.4	Integrin, alpha 4 (antigen CD49D, alpha 4 subunit of VLA-4 receptor)	0.55	**0.033**
*CD28*	NM_006139.3	CD28 molecule	0.46	0.059
*SLAMF7*	NM_021181.3	SLAM family member 7	0.59	0.061

* *p*-value of <0.05 was considered significant. Bold are statistically significat values.

## Data Availability

The data presented in this study are available from the corresponding author upon request.

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
