# Peer review of "Genome-Wide Expression Profile in People with Optic Neuritis Associated with Multiple Sclerosis"

_biomedicines, 2023, doi:10.3390/biomedicines11082209_

Round 1

Reviewer 1 Report

The manuscript analyzes differences in gene expression between patients with optic neuritis and controls. The Supplementary material was not provided so I could not review the whole manuscript properly. Despite of that, the sample size is too small. I consider that it should be increased. Other minor points are:

-        Line 257: “CR3” should be CCR3

-        Table 2: some values are bolded in the p-value column. Is that because they are statistically significant? In that case 0.043 should be bolded too.

-        Table 2: commas should be periods.

-        Data Availability Statement is missing.

Author Response

We appreciate all of the valuable comments from the reviewers of our work. We have revised our manuscript, according to the reviewers’ comments, questions, and suggestions. We believe that the manuscript has been further improved.

Response to Reviewer 1 Comments

We thank you for your time spent carefully reviewing the manuscript, and for your opinions regarding the science and presentation of the material. In what follows the referees’ comments are in black and the authors’ responses are in red.

Comments and Suggestions for Authors

The manuscript analyzes differences in gene expression between patients with optic neuritis and controls. The Supplementary material was not provided so I could not review the whole manuscript properly. Despite of that, the sample size is too small. I consider that it should be increased.

Unfortunately, no supplemental data was attached and you were not able to access all the data. We have now attached it, and we hope you will see all the tables.

Regarding the sample size, we understand that the sample is small, but we still believe that the results obtained are valuable and provide additional information about differences in expression in patients with ON. The small sample size is also described in the manuscript as one of the main limitations of this study, and this was taken into account when interpreting the data. It is important to note that the number of patients with ON as initial presentation and who did not receive immunomodulatory treatment is not large. At the moment it is not possible for us to do this, but it would certainly be good to confirm the results of this study on a larger sample.

Other minor points are:

Line 257: “CR3” should be CCR3

Changed. All abbreviations in the revised manuscript have been checked once more.

Table 2: some values are bolded in the p-value column. Is that because they are statistically significant? In that case 0.043 should be bolded too. Table 2: commas should be periods.

A statement that p-value of <0.05 was considered significant has been added in the table description and 0.043 was also bolded. All commas in this table are now periods.

Data Availability Statement is missing.

Added

Thank you for your very careful review of our paper, and for the comments and suggestions.

Reviewer 2 Report

The manuscript is nicely done and written. The study design is appropriate and apparently, the analyses were carefully performed.  I believe that the results are valuable for the scientific community and has significant scientific merit, as it will probably ignite many further studies in the near future.

However, some points need to be clarified before the publication.

Line 30, 276 – Please explain what CNS stands for?

Line 45 – the citations format should follow the journal guidelines.

Line 63 – please present your hypothesis correctly.

Line 74 – “Borrelia burgdorferi” should be written in italics

Line 81 – please provide ethic approval number.

Line 141 – Did the authors determine primers amplification efficiency?

Line 259 – it is not clear what p-value level was considered as statistically significant.

Line 406 – please correct references style according to the journal guidelines.

Author Response

We appreciate all of the valuable comments from the reviewers of our work. We have revised our manuscript, according to the reviewers’ comments, questions, and suggestions. We believe that the manuscript has been further improved.

Response to Reviewer 2 Comments

We thank you for your time spent carefully reviewing the manuscript, and for your opinions regarding the science and presentation of the material. In what follows the referees’ comments are in black and the authors’ responses are in red.

Comments and Suggestions for Authors

The manuscript is nicely done and written. The study design is appropriate and apparently, the analyses were carefully performed.  I believe that the results are valuable for the scientific community and has significant scientific merit, as it will probably ignite many further studies in the near future.

However, some points need to be clarified before the publication.

Line 30, 276 – Please explain what CNS stands for?

Added. All abbreviations are now explained in the revised manuscript.

Line 45 – the citations format should follow the journal guidelines.

The references style is now aligned with the journal guidelines.

Line 63 – please present your hypothesis correctly.

The hypothesis was changed to: The aim of this study was to determine changes in mRNA expression in blood samples of pwON compared to healthy control subjects.

We hope it is more precise and clear now

Line 74 – “Borrelia burgdorferi” should be written in italics

Changed.

Line 81 – please provide ethic approval number.

The Ethic approval number was mentioned under the Institutional Review Board Statement, but now we added it also in line 82.

Line 141 – Did the authors determine primers amplification efficiency?

Primer amplification efficiency was determined and only primers with efficiency between 90 and 110% were used.

Line 259 – it is not clear what p-value level was considered as statistically significant.

p-value of <0.05 was considered significant. This information is now added in the table description

Line 406 – please correct references style according to the journal guidelines.

The references style is now aligned with the journal guidelines.

Thank you for your very careful review of our paper, and for the comments and suggestions.

Reviewer 3 Report

The authors' aim was to perform a genome-wide expression analysis of whole blood  samples from patients with optic neuritis (ON) and to determine differentially expressed mRNAs compared to healthy control subjects. The study included 8 patients with acute ON and 6 healthy control subjects.

The paper is well written; the abstract is well organized, the introduction includes all necessary explanations, methods are well explained, results well presented and discussion includes all necessary explanations compared to recent literature.

Only one comment:

all abbreviation should be explained in whole words: CYP4F2, MMP-2, MRI, ENA, ANA, ANCA, aCl, SLAM, PTPRC, ITGA4, CCR3, etc.

Author Response

We appreciate all of the valuable comments from the reviewers of our work. We have revised our manuscript, according to the reviewers’ comments, questions, and suggestions. We believe that the manuscript has been further improved.

Response to Reviewer 3 Comments

We thank you for your time spent carefully reviewing the manuscript, and for your opinions regarding the science and presentation of the material. In what follows the referees’ comments are in black and the authors’ responses are in red.

Comments and Suggestions for Authors

The authors' aim was to perform a genome-wide expression analysis of whole blood  samples from patients with optic neuritis (ON) and to determine differentially expressed mRNAs compared to healthy control subjects. The study included 8 patients with acute ON and 6 healthy control subjects.

The paper is well written; the abstract is well organized, the introduction includes all necessary explanations, methods are well explained, results well presented and discussion includes all necessary explanations compared to recent literature.

Only one comment:

all abbreviation should be explained in whole words: CYP4F2, MMP-2, MRI, ENA, ANA, ANCA, aCl, SLAM, PTPRC, ITGA4, CCR3, etc.

Thank you for your very careful review of our paper, and for the comments and suggestions. In the revised paper we explained all abbreviations. All changes are marked in red in the manuscript.